# Caregivers of Children with Autism Spectrum Disorders: The Role of Guilt Sensitivity and Support

**DOI:** 10.3390/jcm13144249

**Published:** 2024-07-20

**Authors:** Amelia Rizzo, Luana Sorrenti, Martina Commendatore, Aurora Mautone, Concettina Caparello, Maria Grazia Maggio, Ahmet Özaslan, Hakan Karaman, Murat Yıldırım, Pina Filippello

**Affiliations:** 1Department of Clinical and Experimental Medicine, University of Messina, 98122 Messina, Italy; luana.sorrenti@unime.it (L.S.); commendatoremartina@icloud.com (M.C.); aurora.mautone.25@gmail.com (A.M.); giuseppa.filippello@unime.it (P.F.); 2Department of Cognitive Sciences, Pedagogical Psychological and Cultural Studies, University of Messina, 98122 Messina, Italy; 3Department of Health Sciences, University Magna Graecia of Catanzaro, 88100 Catanzaro, Italy; concettina.caparello@unicz.it; 4IRCCS Centro Neurolesi “Bonino Pulejo”, 98124 Messina, Italy; mariagrazia.maggio@irccsme.it; 5Child and Adolescent Psychiatry Department, Gazi University Medical Faculty, 06560 Ankara, Turkey; drahmetozaslan@yahoo.com; 6Child Protection Research and Application Center, Gazi University, 06560 Ankara, Turkey; 7Department of Social Work, Faculty of Health Sciences, İstanbul Üniversitesi—Cerrahpaşa, 34320 İstanbul, Turkey; karamannhakann@gmail.com; 8Department of Psychology, Faculty of Science and Letters, Ağrı İbrahim Çeçen University, 04100 Ağrı, Turkey; muratyildirim@agri.edu.tr; 9Department of Social and Educational Sciences, Lebanese American University, Beirut 03797751, Lebanon

**Keywords:** ASD, caregivers, burden, resilience, guilt

## Abstract

**Background/Objectives**: Burden Syndrome, also known as Caregiver Syndrome, particularly affects those who serve in the role of informal caregiver in the presence of family members with conditions. The ABCX dual model examines the impact on the caregiver of the diagnosis of autism spectrum disorder (ASD) on the family. This model considers the severity of the stressor (A), the additional demands of life stress (aA), the family’s internal resources (B), the family’s external resources (bB), the family’s assessment of the situation (C), coping strategies (cC), and outcome (X). The purpose of the present study is to investigate the relationships between resilience, guilt, and burden of care in caregivers of children with ASD. **Methods**: Various assessment instruments were used, including the “Caregiver Burden Inventory” to measure burden, the “Brief Resilience Scale” to assess resilience, the “Guilt Sensitivity Questionnaire” to examine guilt sensitivity, and the “DA.L.I.A.” to collect information on parent and child characteristics. A total of 80 parents/caregivers participated in the research, including 53 women (Age M = 41.72; SD = 7.8) and 27 men (Age M = 43.35; SD = 6.29). **Results**: The findings indicate that individuals’ resilience to stressful events correlates negatively with burden, a developmental subtype. However, guilt seems not to play a significant role in the overall perception of burden. In contrast, it was found that the use of informal supports is associated with higher levels of guilt and emotional burden, whereas the use of formal supports is correlated with higher emotional burden, but not higher perceptions of guilt. **Conclusions**: This study provides important information about the support needed by caregivers and suggests how to address emotional burdens to prevent burnout and support families with children with ASD.

## 1. Introduction

Autism spectrum disorder (ASD) is a neurodevelopmental condition that presents challenges for both the individuals diagnosed and their families. ASD impacts individuals in social communication and interaction, as well as in behavior patterns, interests, and activities, across different areas such as social, educational, and occupational domains [1]. In recent years, there has been a significant rise in the prevalence of ASD. Current estimates show that, in 2016, there were 20 cases per 1000 children aged 8 years, which is almost three times higher than the numbers reported in 2000 and 2002 [2]. This rise, attributed to enhanced diagnostic criteria, genetic factors, and increased awareness, underscores ASD as a growing public health concern, particularly in light of the considerable burden it places on caregivers [3]. Studies have shown that parents of children with ASD often experience increased stress, anxiety, fatigue, and depression due to the challenges of caregiving, accessing support systems, and facing societal biases [4,5,6]. This stress impacts both their health and well-being, as well as their ability to deliver the best care possible. Facing unique challenges highlights the need for customized support services, stressing the significance of recognizing ASD’s frequency and impact on families to meet the community’s needs efficiently.

Parenting a child with ASD has varying effects on mothers and fathers, highlighting the complex nature of family dynamics in these circumstances. Mothers, typically the main caregivers, experience increased stress from their heavy involvement in daily caregiving duties and the struggle with balancing their professional and family lives [7]. Recent studies indicate that fathers also face considerable stress, especially concerning behavioral management and communication challenges linked to ASD, which challenges conventional caregiving roles [8]. A strong social support system is crucial for reducing these strains. This type of support helps parents effectively handle the responsibilities of caregiving. Mothers benefit from supportive networks that offer empathy and understanding, helping them develop coping strategies for dealing with the challenges of raising a child with autism [9]. In contrast, the family unit may face challenges in maintaining marital and familial harmony, potentially resulting in the use of avoidance coping strategies, unless sufficient support systems are available [10]. Fair distribution of parental responsibilities has the potential to improve relationships, emotional resilience, and problem-solving abilities, all of which contribute to a higher quality of life [11]. Sharing facilitates sustainability in caregiving, preventing burnout and ensuring enduring support for children with ASD [12].

The Family Adaptation Model by McCubbin and Patterson (1983) is a theory that identifies coping mechanisms as a central element in the process through which families face and adapt to crises [13]. This model recognizes the following three main categories of resources:-Internal family system resources: the family’s ability to stay united during challenges, maintaining emotional cohesion, adaptability, adequate and accurate communication, and the ability to provide mutual support.-Family members’ resources: the family’s economic situation influences its ability to cope with crises, the health of family members, and the level of education among its members.-Social support provided by external resources outside the family: help from individuals or institutions beyond the family; through the strategies, it is possible to assess a positive response to stress factors.

Resilience and guilt are important factors that significantly impact the experiences of caregivers in this caregiving context [14]. Resilience, the ability to endure and recover from challenges, is crucial in helping caregivers effectively handle the stress and difficulties associated with children with ASD. It illustrates a dynamic process of positive adaptation, essential for preserving psychological well-being when dealing with caregiving burdens [15]. On the other hand, the sense of guilt, which caregivers often experience when fulfilling their responsibilities or seeking help from others, can hinder their ability to provide effective care and take care of themselves. Feelings of guilt may arise from perceived shortcomings in fulfilling the requirements of caring for a child with ASD, or from the need to prioritize caregiving responsibilities over other familial or individual responsibilities [16,17]. 

Several studies have highlighted the role of resilience in addressing the challenges faced by caregivers of children with autism, while the role of guilt as a barrier to accessing help remains largely unexplored. In this study, the overall goal was to investigate the different burdens (social, psychological, work-related, and parental) experienced by caregivers. The specific objectives aimed to explore the following hypotheses:

**Hypothesis** **1** **(H1).***There is a significant correlation between resilience, guilt, and caregiver burden in parents of children with ASD*.

**Hypothesis** **2** **(H2).***Sensitivity to feelings of guilt among caregivers significantly influences their access to formal or informal help*.

The study aimed to determine whether there is a strong relationship between resilience, guilt, and the overall burden experienced by caregivers of children with ASD, as well as to ascertain if the degree of sensitivity to guilt affects their willingness or ability to seek help, whether formal or informal.

## 2. Materials and Methods

### 2.1. Procedure

Participants were recruited through associations, healthcare organizations users, and social networks pages between June and August 2023. They completed the informed consent process before filling out the online questionnaire booklet, which took approximately 20 min in total. Parents independently completed 4 scales in the Italian language. The data collection spanned three months. Participants were each provided with comprehensive guidelines for completing the survey, which included a set of questions and a corresponding sheet for their responses. Their involvement was entirely voluntary, and confidentiality was assured in alignment with the ethical standards outlined in the Declaration of Helsinki for research subjects. Before participating, individuals reviewed and endorsed an informed consent document that detailed the research objectives. This study was non-experimental and designed to avoid any adverse effects on the participants, adhering to the ethical guidelines for online research as set out by the National Committee for Research Ethics in the Social Sciences and the Humanities (NESH), and was approved by the Institutional Review Board.

### 2.2. Measures

#### 2.2.1. Caregiver Burden Inventory (CBI) [18]

This tool is used to assess the caregiving burden experienced by caregivers. It focuses on evaluating various aspects of stress and the burden that caregivers may experience while caring for an individual with an illness or disability. It consists of 24 statements divided into five sections, each representing a specific aspect of the caregiving burden. Each section comprises five statements reflecting different aspects of that particular dimension of the caregiving burden. Caregivers respond to each statement by assigning a score that reflects the degree to which the statement reflects their experience, using a scale from 0 to 4 (0 = Not at all, 1 = A little, 2 = Moderately, 3 = Quite a bit, and 4 = Extremely). The total sum of scores across different sections indicates the overall caregiving burden for the caregiver. The sections include the following:Time-dependence burden (items 1–5—denoted by the letter T) provides an overview of the caregiving burden (level of daily commitment, impact on personal freedoms, etc.) associated with the patient’s dependence on the caregiver. For example: “My family member depends on me”.Evolutionary burden (items 6–10—denoted by the letter S) refers to the impact of caregiving on the parent’s social and personal life, as well as their expectations regarding life. For example: “I would have expected something different at this point in my life”.Physical burden (items 11–14—denoted by the letter F) reflects feelings of chronic fatigue and somatic health problems that caregivers may experience due to caregiving responsibilities. For example: “Caring for him/her has made me physically weaker”.Social burden (items 15–19—denoted by the letter D) focuses on the caregiver’s perception of role conflict and social relationships influenced by caregiving responsibilities. For example: “I feel resentment toward my family members who could help but do not”.Emotional burden (items 20–24—denoted by the letter E) concentrates on the caregiver’s feelings toward the patient, which can be influenced by the patient’s unpredictable or bizarre behaviors. For example: “I feel ashamed of him/her”.

#### 2.2.2. Brief Resilience Scale (BRS) [19]

The Brief Resilience Scale was developed to measure a unified construct of resilience in response to stress, particularly the ability to recover from stress. Specifically, it comprises 6 items. Items 1, 3, and 5 are formulated positively, while points 2, 4, and 6 are formulated negatively. The BRS is assessed by reverse-coded items 2, 4, and 6, and by calculating the mean of all six items. Response options include a 5-point scale: 1 = strongly disagree, 2 = disagree, 3 = neutral, 4 = agree, and 5 = strongly agree.

#### 2.2.3. Guilt Sensitivity Questionnaire (GSQ) [20]

The Italian version of the Guilt Sensitivity Questionnaire is a self-report questionnaire that measures vulnerability to feelings of guilt, specifically beliefs regarding the emotion of guilt itself as being dangerous, unsustainable, severe, and threatening. Based on several considerations, the authors developed the Guilt Sensibility Scale consisting of 10 items. This scale assesses subjective sensitivity to guilt by analyzing the following: the tendency to avoid this feeling, its influence on the participant’s life, and their ability to tolerate it. The main objective of this study was to test the psychometric properties of this scale. To achieve this, the scale was administered to a sample of 916 participants. In terms of psychometric properties, the instrument appears to exhibit significant and promising reliability and validity.

#### 2.2.4. DA.L.I.A. (2012) [21]

The Caregiver Information Questionnaire was developed as part of the DA.L.I.A. caregiver support project, supported by the Ministry for Equal Opportunities of the Emilia-Romagna Region. It is designed to gather detailed information from caregivers of family members with disabilities. It consists of 29 items covering demographic information, caregiving frequency and intensity, motivations for caregiving, and knowledge and use of educational, psychological, and social support.

The following is a detailed description of the questionnaire sections: 1: Personal Information; 2: Socio-economic Information; 3: Disability Details; 4: Assistance and Care; 5: Motivations and Evaluations; 6: Care Burden and Support; 7: Impact on the User; 8: Support and Services; 9: Financial Impact; 10: Service Selection Priorities; 11: Work and Care Balance; 12: Space for Comments. The questionnaire can be accessed at the following link: http://dalia.anzianienonsolo.it/?page_id=59, accessed on 18 December 2022.

In the present study, the reliability of the instruments used was assessed using Cronbach’s alpha to ensure the internal consistency of the questionnaires in their Italian version. The Caregiver Burden Inventory (CBI) demonstrated high reliability across its subscales, with Cronbach’s alpha values of 0.951 for time-dependence burden, 0.948 for evolutionary burden, 0.935 for physical burden, 0.937 for social burden, and 0.934 for emotional burden. The Brief Resilience Scale (BRS) also showed good reliability with a Cronbach’s alpha of 0.831. Additionally, the Guilt Sensitivity Questionnaire (GSQ) exhibited excellent reliability, with a Cronbach’s alpha of 0.947.

### 2.3. Participants

The sample size was calculated using a sample size calculator, inputting a confidence level of 95%, a margin of error of 5%, and a population size of 78,826 (total children diagnosed with autism in Italy) [11]. The default population proportion of 50% was used, as the actual proportion was unknown, ensuring a conservative and adequate sample size. The calculation resulted in a sample size of 66, indicating that this number of measurements or surveys is necessary to achieve the desired statistical accuracy and confidence. From an initial sample of 120 cases, 80 were included in the final analysis as valid protocols. Inclusion criteria were (1) assisting a child with relevant diagnoses related to autism or related disorders; (2) consenting to anonymously complete the questionnaire; (3) having internet access and a sufficient understanding of the Italian language; (4) being free from diagnoses. For this reason, 40 protocols of caregivers assisting children with unrelated diagnoses—such as ADHD, Klinefelter syndrome, cancers, strokes, and different neurological disorders—were excluded (See Figure 1).

The research involved a final sample of 80 parents/caregivers, comprising 53 women and 27 men. More than half of them reside in the islands, followed by those residing in the north and south, and, finally, a small portion in the central region. The age range for female parents varied from 26 to 62, with an average age of 41.72 years and an SD = 7.08. For male parents, the age range was from 30 to 53, with an average age of 43.35 years and an SD = 6.29. The majority are full-time employees, followed by self-employed individuals, part-time employees, unemployed, and jobless (See Table 1).

## 3. Results

The data analysis was conducted using SPSS version 27.0 (SPSS Inc., Chicago, IL, USA) with a significance level set at *p* < 0.05. Descriptive statistics were presented as mean ± standard deviation, while categorical variables were expressed as frequencies and percentages. Correlations between continuous variables were assessed using the Pearson correlation coefficient. For group comparisons, Student’s *t*-test for independent samples was employed.

RQ1: Is there a significant correlation between resilience, guilt sensitivity, and caregiver burden in parents of children with autism spectrum disorder (ASD)?

### 3.1. Correlation Analysis

The correlation table (Table 2) illustrates the relationships among various dimensions of perceived burden, resilience, as well as guilt sensitivity.

The “Brief Resilience” shows no significant correlations with most variables, but exhibits a moderate negative correlation with “Evolutionary Burden”. This suggests that individuals with higher resilience might perceive fewer burdens related to evolutionary expectations, such as challenges linked to adaptation and change over the lifespan. Conversely, “Guilt Sensitivity” does not display significant correlations.

Correlations among “Objective Burden”, “Evolutionary Burden”, “Physical Burden”, “Social Burden”, and “Emotional Burden” are all positive and significant, indicating that these burden aspects tend to increase together. Particularly, “Evolutionary Burden” and “Physical Burden” exhibit the strongest correlation, implying a strong connection between evolutionary challenges and physical difficulties. “Social Burden” shows robust correlations with “Physical Burden” and “Evolutionary Burden”, suggesting that challenges in social relationships might be interconnected with physical issues and adaptation to change.

Figure 2 shows the co-occurrence of psychological symptoms of burden among caregivers.

Notably, symptoms of excessive fatigue and insomnia have the highest co-occurrence rate at 46.16%, suggesting a strong link between physical exhaustion and sleep disturbances. Crying fits and symptoms of excessive fatigue also show a high co-occurrence rate of 42.47%, indicating that emotional distress often accompanies physical tiredness. The other symptom pairs, such as outbursts of anger with crying fits, insomnia, or excessive fatigue, and crying fits with insomnia, exhibit moderate to lower co-occurrence rates, reflecting varying degrees of association between these symptoms.
RQ2: Does guilt sensitivity among caregivers significantly impact their access to formal or informal help?

### 3.2. Request for Help and Guilt

Among caregivers of children with autism, the use of informal assistance has been associated with a higher sense of guilt and increased emotional burden. Analysis of mean differences using the independent samples *t*-test revealed significant differences in emotional burden [t = −2.82; df(78); *p* = 0.007] and guilt sensitivity [t = −3.38; df(78); *p* = 0.001] between those who do and do not utilize help in caring for the child, with higher levels observed in the former group (Table 3).

Conversely, when comparing caregiver parents who use a day center/formal assistance, a higher emotional burden is evident [t = −2.37; df(78); *p* = 0.02], but not guilt (Table 4). The results indicate a statistically significant difference for both those who seek assistance and those who do not.

## 4. Discussion

The primary aim of this study was to explore the various burdens (social, psychological, occupational, and parental) experienced by caregivers, with a specific focus on caregivers of children with ASD. The following two main hypotheses guided the research: the first examined whether there was a significant correlation between caregivers’ resilience, their sense of guilt, and the overall burden they faced. The second hypothesis investigated the impact of caregivers’ guilt sensitivity on their recourse to both formal and informal assistance. In summary, the study aimed to analyze the link between caregivers’ emotional and psychological factors and their strategies for managing the burden associated with caring for children with ASD.

The results pertaining to the first hypothesis offer interesting insights. The lack of a direct relationship between “Brief Resilience” and most perceived burden variables indicates that resilience, in general, does not seem to directly influence how caregivers perceive most of the burdens associated with their activity [22,23]. However, the moderate negative correlation with “Evolutionary Burden” is noteworthy: it suggests that higher resilience might indeed help caregivers perceive challenges related to adaptation and change over the lifespan less heavily. This could have significant implications for caregiver training and support, emphasizing the importance of developing resilience to better manage evolutionary changes.

The positive and significant correlations among various aspects of burden (objective, evolutionary, physical, social, and emotional) suggest that these elements tend to increase concurrently. This means that, if a caregiver experiences an increase in one dimension of burden, they are likely to encounter an increase in others as well. In particular, the strong correlation between “Evolutionary Burden” and “Physical Burden” might indicate that challenges related to change and adaptation are closely linked to physical difficulties, as found in other studies [24]. Additionally, the correlation between “Social Burden” and “Physical Burden” or “Evolutionary Burden” suggests that issues in social relationships may be intrinsically connected to physical challenges and those related to change.

Similarly, Stuart and McGrew (2009) highlighted family stress associated with the initial diagnosis of ASD, emphasizing that parents experience stress in parallel with other life stressors [25]. This aligns with the results of the present study showing a correlation among various aspects of perceived burden. Several studies exploring caregiver burden in childhood neurodevelopmental disorders such as intellectual disability [26], attention deficit hyperactivity disorder [27], and Tourette syndrome [28] have reported that caregiver burden increases with the severity of symptoms and the presence of concomitant conditions.

The second finding highlights how, in caregivers of children with autism, the use of informal assistance is associated with a greater sense of guilt and higher emotional burden. The analysis revealed statistically significant differences in the levels of emotional burden and guilt sensitivity between caregivers who seek help in caring for the child and those who do not, showing higher emotional burden and guilt in the former group. Moreover, when comparing parents/caregivers who utilize a daytime center or other formal assistance, a higher emotional burden was observed in this group, although no significant differences were found regarding guilt. This suggests that, while the use of both formal and informal assistance is linked to increased emotional burden, only the use of informal assistance is associated with heightened guilt. In summary, the results indicate that both informal and formal assistance usage is associated with greater emotional burden among caregivers of children with autism, yet the sense of guilt is more pronounced among those who resort to informal aid.

The study by Fong, Gardiner, and Iarocci (2020) highlighted how formal support could enhance the resilience of families with children with ASD [29]. It was observed that families benefiting from such support tend to express greater satisfaction with external guidelines and assistance received. This satisfaction, in turn, correlates with higher resilience in families [30]. In essence, formal social supports prove to be a fundamental resource in strengthening families’ ability to adapt and positively respond to challenges associated with ASD.

Guilt, as a moral emotion, along with self-awareness, can provoke negative self-assessments and feelings of distress related to perceived failures or transgressions [31,32,33]. Lewis (1971) distinguishes guilt from shame, noting that, while guilt leads to tension, remorse, and regret for wrongdoing, shame induces feelings of worthlessness and a desire to hide [34]. Other studies link a predisposition to shame to avoidance tendencies [35] and psychopathological symptoms such as somatization, depression, anxiety, or hostility [36,37]. Parents of neurotypical children may experience guilt and shame, but these experiences might differ among parents of children with ASD, given that the behavior of the latter is often socially misunderstood [38]. Moreover, many studies indicate that guilt is negatively correlated with self-forgiveness in both groups of parents [39,40]. This might be because strong emotional responses like remorse, tension, or regret can hinder self-forgiveness [41]. Therefore, the increase in guilt and shame can lead to increased parental stress, both in parents of neurotypical children and in those with children with ASD. Caregivers who resort to informal help show a higher sense of guilt, possibly linked to feelings of inefficacy and the perception of causing further issues within the family. This can be an additional risk factor burdening the parent.

In the context of caregiving for children with ASD, the simultaneous presence of symptoms such as ‘Outbursts of anger’ and ‘Crying fits’, showing the strongest correlation (42.47), might suggest a great deal of stress and emotional burden felt by caregivers. The strong correlation indicates that caregivers may be reacting to chronic stressors with various emotional responses, reflecting the demanding environment they frequently experience. There is an important relationship between ‘Symptoms of excessive fatigue’ and ‘Insomnia’ (32.5), which might point to the widespread sleep problems faced by caregivers. This may aggravate daytime fatigue and influence their caregiving quality. Experiencing a range of strong emotions due to having a child with ASD and feeling overwhelmed by the responsibilities of caring for these children is in line with findings from previous research [42,43]. The emotional responses cover senses of anxiety due to uncertainty, sadness and guilt stemming from the loss, or anger arising from difficulty in accepting the situation. These emotions may contribute to the development of problems with both mental and physical health [44,45]. These unpleasant conditions emphasize the importance of support systems that understand the complex challenges faced by caregivers of children with ASD [46]. Focusing on emotional regulation and sleep quality in caregiver support programs is essential for reducing the overall burden. It is evident that any intervention designed to support caregivers should take a comprehensive approach, recognizing the close relationship between emotional and physical well-being in that case. This study illustrates the need for customized interventions targeting the linked symptoms experienced by caregivers. Interventions may include stress management, emotional coping strategies, and practical support for problems with sleep. Healthcare providers can improve caregiver support by identifying symptom patterns, leading to better quality of life and caregiving capacities [47]. Further study is needed to fully understand the causal relationships between these simultaneous symptoms and to develop specific strategies for dealing with them, since recent studies demonstrated that a caregiver’s burden is reversible [48,49]. Exploring how symptoms grow over time and affect caregivers could help to develop evidence-based support services for this group.

## 5. Limitations

This study provides valuable insights into the experiences of caregivers for children with ASD, specifically examining the relationship between resilience, sense of guilt, and different types of burdens. Nevertheless, it is important to consider various limitations. The use of a relatively small and potentially non-diverse sample limits the ability to generalize the findings to larger populations. The study’s participants may not adequately reflect the wide variety of caregivers’ backgrounds, including variations in culture, socio-economic status, and the severity of the children’s ASD. The comparison of caregiver stress levels across different cultural contexts reveals significant societal and cultural influences on their experiences, which limits the generalizability of the results to other contexts, despite the similarities found. A study conducted in the United States found that caregivers face high levels of stress linked to the severity of their child’s ASD symptoms and inconsistent social support systems [50]. Conversely, research in Japan and Australia indicates that societal expectations and stigma surrounding ASD contribute to heightened stress and reluctance to seek help among caregivers [51]. In India, Gupta et al. [52] demonstrated that caregivers experience substantial burdens due to limited resources and societal acceptance, often relying heavily on informal support networks. These findings underscore the importance of culturally sensitive interventions and support systems tailored to meet the unique needs of caregivers in diverse contexts.

Furthermore, the employing of a cross-sectional design in this study restricts our ability to determine causality or the direction of the relationships between the variables under investigation. Adopting a longitudinal approach would facilitate a deeper understanding of the growth of these dynamics over time. A further limitation comes from relying solely on self-reported data, which could lead to bias, as participants may intentionally provide inaccurate or exaggerated information due to social desirability or memory-related difficulties. In addition, the study provides no insight into the quality, accessibility, or types of care (casual or structured) provided by caregivers, which could significantly impact their experiences and perceived challenges.

Finally, the study fails to consider additional psychological and emotional factors, such as anxiety, depression, and coping strategies, that may have an impact on caregiver experiences, despite the study’s focus on resilience and sense of guilt. To address these limitations, future studies should utilize a greater variety and larger samples, longitudinal study designs, different ways to gather data, and a wider range of psychological parameters. This may contribute to improving the knowledge and support offered to caregivers of children with ASD.

## 6. Conclusions

The study revealed the complex relationship among resilience, guilt, and different burdens faced by caregivers of children with ASD. The relationship between informal support and feelings of guilt, as well as the correlation between formal support and resilience, underscores the importance of support systems in managing the difficulties of ASD caregiving. The study highlights the importance of having thorough support systems that target emotional regulation and practical caregiving challenges. This research provides to the growing body of studies aiming to enhance the well-being of caregivers and their children by promoting specific interventions which promote caregiver resilience and alleviate the emotional burdens of caregiving.

## Figures and Tables

**Figure 1 jcm-13-04249-f001:**
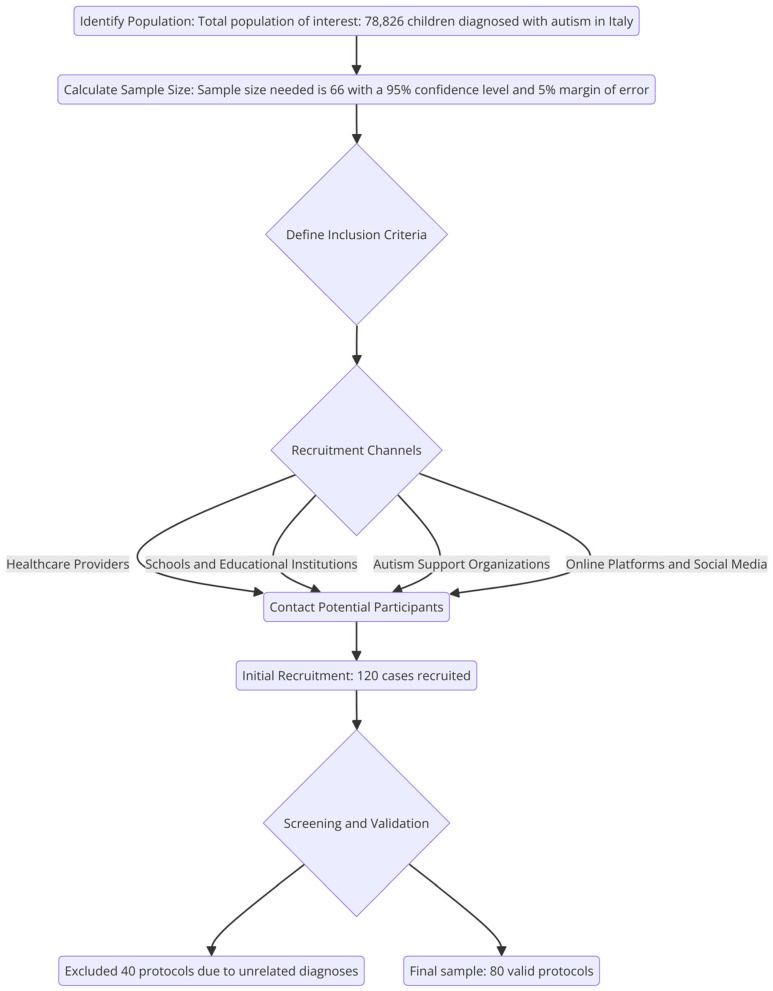
Detailed recruitment procedure flowchart.

**Figure 2 jcm-13-04249-f002:**
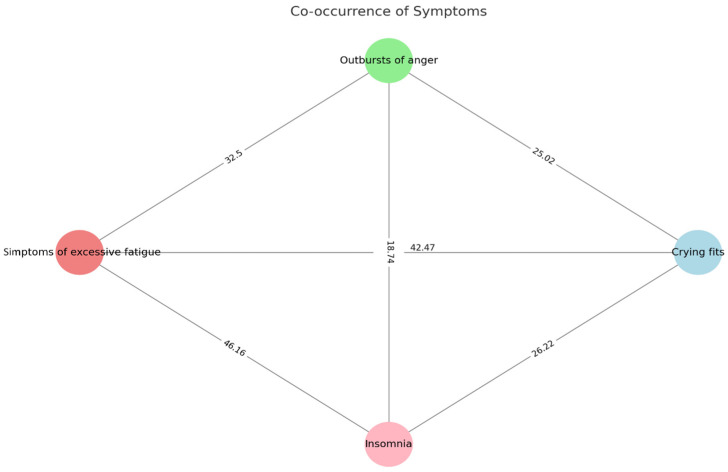
Co-occurrence of caregiver burden symptoms. Note: In the graph, each circle (node) represents a symptom. Each line (edge) connects two symptoms that have occurred together. The number on the line indicates the percentage with which those two symptoms were reported together.

**Table 1 jcm-13-04249-t001:** Descriptive characteristics of the caregivers sample (N = 80).

How many children do you have?	%
1 child	47.3
2 children	36.3
3 children	12.5
More than 3 children	3.9
How many have disabilities?	
One child with a disability	85
Two children with a disability	12.5
Three children with a disability	2.5
What is your employment status?	
Self-employed	25
Full-time employed	35
Part-time employed	16.3
Not employed	23.7
What is your income bracket?	
from EUR 32,000.01 to EUR 40,000.00	2.5
from EUR 40,000.01 to EUR 48,000.00	1.3
from EUR 16,000.01 to EUR 24,000.00	37.4
from EUR 24,000.01 to EUR 32,000.00	2.5
from EUR 8000.01 to EUR 16,000.00	41.3
up to EUR 8000.00	15
Where do you reside?	
Center	5
Isles	65
North	15
South	15
What type of disability does the person you assist have?	
Intellectual/Cognitive	81
Physical/Motor	10
Both	9
How many hours a day are you engaged in caregiving?	
Less than two hours	2.5
From three to six hours	22.5
More than six hours	75
The level of dependency of the family member you assist is:	
Severe	43.8
Moderate	42.5
Mild	12.5
None	1.3
With the assisted person:	
Lives in the same city	1.3
Lives nearby	1.3
Co-lives in the same house	97.5
How long have they been taking care of their family member?	
For over five years	75
For over three years	15
For over a year	5
Since birth	5
What motivated you to take care of your family member?	
Out of love and affection	87.57
Due to responsibility and a sense of duty	9.42
Because I had no other choice: no one else could take care of them	3.01
How often does your caregiving activity occur?	
At least once every fifteen days	1.3
Multiple times a week	6.3
Daily	90
Once a week	2.5
On average, how many hours per week do you dedicate to caregiving activities?	
Up to 20 h	8.8
Up to 40 h	20
Up to 70 h	31.3
Over 70 h	40
Do you feel you need help?	
Sometimes	32.5
Frequently	50
Never	3.8
Very frequently	13.8
How do you rate the burden of caregiving activities for your family member?	
Light	5
Very heavy	11.3
Heavy	51.2
Manageable	32.5
Have you recently experienced:	
Crying spells	20.79
Outbursts of anger	16.68
Symptoms of excessive tiredness	40.38
Insomnia	22.46
Do you believe caregiving has deteriorated your quality of life and social relationships?	
Partially	52.5
Not at all	6.3
Significantly	41.3
Do you seek assistance in carrying out caregiving activities?	
No	25
Yes	75
Do you rely on informal help?	
Family member	85.8
Babysitter	0.65
Untrained caregiver	0.65
Neighbors	0.65
Volunteers	3.75
None	7.6
“Autism Space”	1.3
Do you attend a day center?	
No	52.5
Yes	47.5
How much does the cost of caring for your family member affect your income?	
Lightly	23.8
Very heavily	15
Heavily	61.3

**Table 2 jcm-13-04249-t002:** Correlations among caregiver burden, resilience, and guilt sensitivity.

	1	2	3	4	5	6
1. Brief Resilience	1					
2. Guilt Sensitivity	0.055	1				
3. Time Dependence B.	−0.009	0.009	1			
4. Developmental B.	−0.293 **	−0.005	0.625 **	1		
5. Physical B.	−0.203	0.010	0.783 **	0.819 **	1	
6. Social B.	−0.156	−0.079	0.687 **	0.766 **	0.795 **	1
7. Emotional B.	−0.174	0.206	0.561 **	0.765 **	0.685 **	0.712 **

Note: ** Correlation is significant at the 0.01 level (2-tailed). B. = burden.

**Table 3 jcm-13-04249-t003:** Comparison between parents who use/do not use informal assistance in caregiving.

	Yes (N = 60)		No (N = 20)		*p* Value for the Difference	
Measures	Mean	SD	Mean	SD	T	Sig.
Time Dependence B.	13.50	5.20	14.35	4.22	−0.66	0.11
Developmental B.	9.80	6.23	11.00	6.08	−0.74	0.71
Physical B.	6.33	3.41	7.70	3.21	−1.57	0.94
Social B.	8.28	6.74	10.65	4.80	−1.45	0.03
Emotional B.	8.03	6.19	4.70	3.88	2.25	0.01
Guilt Sensitivity	28.60	6.11	22.30	9.86	3.38	<0.001
Brief Resilience	2.99	0.42	3.00	0.66	−0.06	0.01

Legend: B. = burden.

**Table 4 jcm-13-04249-t004:** Comparison between parents who utilize/do not utilize formal assistance in caregiving.

	Yes (N = 38)		No (N = 42)		*p* Value for the Difference	
Measures	Mean	SD	Mean	SD	T	Sig.
Time Dependence B.	13.50	5.20	14.35	4.22	−0.66	0.10
Developmental B.	9.80	6.23	11.00	6.08	−0.74	<0.001
Physical B.	6.33	3.41	7.70	3.21	−1.57	0.07
Social B.	8.28	6.74	10.65	4.80	−1.45	0.01
Emotional B.	8.03	6.19	4.70	3.88	2.25	<0.001
Guilt Sensitivity	28.60	6.11	22.30	9.86	3.38	0.01
Brief Resilience	2.99	0.42	3.00	0.66	−0.06	0.01

Legend: B. = burden.

## Data Availability

The raw data supporting the conclusions of this article will be made available by the corresponding author on reasonable request.

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
