# Peer review of "Caregivers of Children with Autism Spectrum Disorders: The Role of Guilt Sensitivity and Support"

_jcm, 2024, doi:10.3390/jcm13144249_

Round 1
Reviewer 1 Report
Comments and Suggestions for Authors
This article investigates the relationships between resilience, guilt, and burden of care in Caregivers of children with ASD. Some points for a major revision can be found below:
Please justify the sample size. Was a software used to estimate the sample size? Is it representative?
Please describe in detail the recruitment procedure. It is not clear.
Please add Cronbach's alphas for all administered questionnaires.
Authors can use in the discussion relevant references from other disorders diagnosed in children and how policymakers and caregivers are affected (please read and discuss a relevant recent article https://www.mdpi.com/2036-7503/16/2/36). This article could also assist authors in the future implications of their findings and how to translate them into policy.
Did the caregivers demonstrate any cognitive deficits? Or were they free from diagnoses? Please mention in a clear way.
Finally, data from other cultural contexts could be added in the discussion section about the examined variables in this special population,
Comments on the Quality of English LanguageModerate English language editing.
Author Response
Response Letter
Dear Editor
We appreciate the thorough reviews of our manuscript titled "Caregivers of Children with Autism Spectrum Disorders: The Role of Guilt Sensitivity and Support". Coauthors and I are particularly grateful for the ameliorating suggestion and we are more than satisfied of the improvements.
We have addressed each comment and suggestion provided by the reviewers and have revised the manuscript accordingly. Below are our detailed responses to the points raised:
REVIEWER 1
Sample Size Justification:
Comment: Please justify the sample size. Was software used to estimate the sample size? Is it representative?
Response: We have now included a detailed justification of the sample size in the Methods section. We used a calculator software to estimate the sample size required for our study to ensure it is representative of the population (95% of CI).
Recruitment Procedure:
Comment: Please describe in detail the recruitment procedure. It is not clear.
Response: We have elaborated on the recruitment procedure in the Methods section, specifying the inclusion criteria and the steps taken to recruit participants, and an explicative flowchart was added.
Cronbach's Alphas:
Comment: Please add Cronbach's alphas for all administered questionnaires.
Response: The Cronbach’s alpha values for all administered questionnaires have been added to the manuscript, ensuring the reliability of the scales used.
Here are the values:
Caregiver Burden Inventory (CBI): Time-dependence burden (.951), Evolutionary burden (.948), Physical burden (.935), Social burden (.937), Emotional burden (.934)
Brief Resilience Scale (BRS): (.831)
Guilt Sensitivity Questionnaire (GSQ): (.947)
Additional References:
Comment: Authors can use in the discussion relevant references from other disorders diagnosed in children and how policymakers and caregivers are affected (please read and discuss a relevant recent article https://www.mdpi.com/2036-7503/16/2/36). This article could also assist authors in the future implications of their findings and how to translate them into policy.
Response: We have integrated relevant references from other disorders diagnosed in children into the discussion section, including the suggested article to enhance the context and implications of our findings.
Cognitive Deficits in Caregivers:
Comment: Did the caregivers demonstrate any cognitive deficits? Or were they free from diagnoses? Please mention in a clear way.
Response: We have clarified the cognitive status of the caregivers in the Participants section, stating whether any cognitive deficits were present or if they were free from diagnoses.
Cultural Contexts:
Comment: Finally, data from other cultural contexts could be added in the discussion section about the examined variables in this special population.
Response: We have expanded the discussion section to include data from other cultural contexts, providing a comprehensive comparison and highlighting how cultural and societal differences influence caregiver experiences.
REVIEWER 2
Pilot Study Notation:
Comment: Due to the small size of the study group, I propose to note that these are pilot studies (in the title and content of the manuscript).
Response: We have noted that this study has limited generalizability in the content of the manuscript and in the limitation section. However, sample size calculation revealed sample adequacy with respect to the reference population.
Reference Formatting:
Comment: Authors should write references in the text in a different way (in square brackets with the appropriate number).
Response: References in the text have been reformatted to use square brackets with the appropriate number.
Data Collection Method:
Comment: Lines 121-122: Were all questionnaires collected online?
Response: We have clarified in the Methods section that all questionnaires were collected online.
Statistical Methods and Software:
Comment: The authors did not specify what statistical methods (and what program) they used for the calculations.
Response: We have specified the statistical methods and the software (SPSS 27.0) used for our calculations in the Methods section. Thank you for noticing.
Missing Legend:
Comment: The legend is missing under table 4.
Response: A legend has been added under Table 4. Thank you for noticing.
Harmonizing Statistical Significance:
Comment: Please harmonize the recording of statistical significance in table 3 and table 4.
Response: We have harmonized the recording of statistical significance across Table 3 and Table 4 for consistency. Thank you for noticing.
Inclusion Flowchart:
Comment: The results lack an all-encompassing table or a flowchart for the selection of patients included with the step-by-step reasons for exclusion.
Response: A flowchart detailing the selection of participants and reasons for exclusion has been added to the manuscript. This has been a brilliant idea and suggestion, really many thanks.
Reference Order:
Comment: References should not be given alphabetically but in the order they appear in the text.
Response: References have been reordered to follow the sequence in which they appear in the text.
We hope that these revisions meet the expectations of the reviewers and improve the quality of our manuscript. Thank you for your valuable feedback.
Sincerely,
The corresponding author on behalf of all co-authors
Reviewer 2 Report
Comments and Suggestions for Authors
Thank you for the opportunity to review this paper. This is a timely and important issue to explore. The aim of the study was to is to investigate the relationships between resilience, guilt, and burden of care in Caregivers of children with autism spectrum disorder. It is mostly clear, there is, however, issues that must be resolved.
Several comments and suggestions for the authors.
- Due to the small size of the study group, I propose to note that these are pilot studies (in the title and content of the manuscript).
- Authors should write references in the text in a different way (in square brackets with the appropriate number).
- Lines 121-122: Were all questionnaires collected online?
- Can the authors provide the Cronbach’s alpha value for all questionnaires?
- Was the study group representative of the population? How was the size of the study group planned? Which formula was used to calculate the sample size?
- The authors did not specify what statistical methods (and what program) they used for the calculations.
- The legend is missing under table 4.
- Please harmonize the recording of statistical significance in table 3 and table 4.
- The results lack an all-encompassing table, a flowchart for the selection of patients included with the step-by-step reasons for exclusion.
- References should not be given alphabetically, but in the order they appear in the text.
Author Response

(The authors gave the same response as above.)

Round 2
Reviewer 2 Report
Comments and Suggestions for Authors
I would like to thank the authors for the corrections in accordance with the review. I have no further comments.